# Functional investigation suggests *CNTNAP5* involvement in glaucomatous neurodegeneration obtained from a GWAS in primary angle closure glaucoma

Sudipta Chakraborty [1,2], Jyotishman Sarma[1,2], Shantanu Saha Roy[1], Sukanya Mitra[1,2], Sayani Bagchi[1], Sankhadip Das [1], Sreemoyee Saha[1], Surajit Mahapatra[1], Samsiddhi Bhattacharjee [1,2]*, Mahua Maulik [1]*, Moulinath Acharya [1,2]*

**1** Biotechnology Research and Innovation Council-National Institute of Biomedical Genomics (BRIC-NIBMG), Kalyani, India, **2** Regional Centre for Biotechnology, Faridabad, India

* sb1@nibmg.ac.in (SB); mm1@nibmg.ac.in (MM); ma1@nibmg.ac.in (MA)

## Abstract

Primary angle closure glaucoma (PACG) affects more than 20 million people worldwide, with an increased prevalence in south-east Asia. In a prior haplotype-based Genome Wide Association Study (GWAS), we identified a novel *CNTNAP5* genic region, significantly associated with PACG. In the current study, we have extended our perception of *CNTNAP5* involvement in glaucomatous neurodegeneration in a zebrafish model, through investigating phenotypic consequences pertinent to retinal degeneration upon knockdown of *cntnap5* by translation-blocking morpholinos. While *cntnap5* knockdown was successfully validated using an antibody, immunofluorescence followed by western blot analyses in cntnap5-morphant (MO) zebrafish revealed increased expression of acetylated tubulin indicative of perturbed cytoarchitecture of retinal layers. Moreover, significant loss of Nissl substance is observed in the neuro-retinal layers of cntnap5-MO zebrafish eye, indicating neurodegeneration. Additionally, in spontaneous movement behavioural analysis, cntnap5-MO zebrafish have a significantly lower average distance traversed in light phase compared to mismatch-controls, whereas no significant difference was observed in the dark phase, corroborating with vision loss in the cntnap5-MO zebrafish. This study provides the first direct functional evidence of a putative role of *CNTNAP5* in visual neurodegeneration.

## Author summary

Glaucoma is the most common irreversible blindness worldwide where neurodegeneration manifests in the retina and optic nerve head. Primary angle closure glaucoma (PACG) is one of the major subsets of glaucoma, prevalent mostly in east and south-east Asia. In the current study we report a putative candidate *CNTNAP5*, a neurexin family gene, obtained from our previously published GWAS on PACG, involved in glaucomatous neurodegeneration. We have used antisense oligo mediated knockdown of the

**Data Availability Statement:** All data are available in the Figure and Supplementary data files contained in the manuscript.

**Funding:** This work was supported mostly by NIBMG intramural funds to MA and an extramural research grant from the Department of Biotechnology, Government of India (BT/PR11558/MED/12/649/2014) to MA. SC is a recipient of the senior research fellowship from the Indian Council of Medical Research (ICMR) (No.3/1/3(8)/OPH/2020-NCD-II). The funders had no role in study design, data collection and analysis, decision to publish, or preparation of the manuscript.

**Competing interests:** The authors have declared that no competing interests exist.

human *CNTNAP5* ortholog *cntnap5a* and *cntnap5b* in the zebrafish. We have observed disrupted retinal cytoarchitecture with higher expression of acetylated tubulin and increased chromatolysis and cell death in zebrafish larvae injected with antisense oligos against *cntnap5a* and *5b* at 4 days post-fertilization when compared with the mismatch control oligo injected zebrafish larvae, pointing towards neurodegeneration. Moreover, the double knockdown zebrafish larvae showed restricted movement in the light phase only in a simple set-up with software and camera to record and analyse the movement of zebrafish larvae. This phenomenon further validates the loss of vision in zebrafish larvae lacking both *cntnap5* orthologs. Altogether, this study, for the first time, provides direct evidence of CNTNAP5 involvement in neuronal loss in the retina and glaucoma.

## Introduction

Primary angle closure glaucoma (PACG) is a major blinding eye disease, affecting over 20 million patients globally, more prevalent in east and south-east Asia [1]. It is characterized by progressive closure of the anterior chamber angle, resulting in blocked drainage of aqueous humour and increased intraocular pressure, which ultimately damages the optic nerve [2,3]. Eventually, loss of retinal layers and the degeneration of the optic nerve can result in irreversible vision loss, leading to blindness [4]. The disease has a strong heritable component, suggesting genetic factors play an important role in its pathogenesis [5].

A few Genome Wide Association Study (GWAS) in Asian populations revealed associations between PACG and genetic variants in regions near or within the genes *PLEKHA7*, *COL11A1*, and *PCMTD1-ST18* [6]. Later, 5 new loci have been identified to be associated with disease risk near the genes *DPM2-FAM102A*, *FERMT2*, *GLIS3* and *CHAT* [7]. Notably, all these genes have known roles in ocular development, suggesting shared genetic mechanisms in angle development and closure. However, genomic factors contributing towards visual neurodegeneration aspect of the disease remains unexplored and elusive.

Previously, our extreme phenotype haplotype-based genome-wide association study identified *CNTNAP5* genic region as novel genomic loci to be significantly associated with PACG and one of its major endophenotype, cup-to-disc ratio (CDR) [8]. An increased CDR, which reflects the relative enlargement of the optic cup compared to the disc, can indicate the loss of retinal ganglion cells and thinning of the neuro-retinal rim, making it useful for monitoring patients with glaucomatous neurodegeneration. Subsequently, our *in silico* analytical data further provided additional lines of evidence suggesting that *CNTNAP5* is implicated in potential expression changes and altered regulatory gene networks in glaucomatous pathophysiology [8]. CNTNAP5 (Contactin Associated Protein-5) is a neural transmembrane protein, mainly enriched in myelinated axons that belongs to the neurexin superfamily [9]. It is a class of cell adhesion molecule of contactin family, found primarily in the brain, where they play critical roles in neurite formation, neuronal development, axonal domain organisation, and axonal guidance [10]. Their absence causes axon malformation and poor nerve transmission. Earlier reports detected possible connection between rare variants in *CNTNAP5* with neurodevelopmental disorders or neurological diseases [11]. However, functional evidence validating the putative role of *CNTNAP5* in glaucomatous neurodegeneration remains to be determined.

There are numerous examples in the literature of GWAS loci that fail to be corroborated in functional studies, underscoring the need for rigorous follow-up analyses [12]. Moreover, translating GWAS findings into biological insights remains challenging, as the majority of disease-associated variants map to non-coding regions with unclear functional effects [13]. In

this study, we have extended the current understanding of the molecular contributions of *CNTNAP5* to glaucomatous neurodegeneration. To incorporate genomic annotation data and emphasize role of *CNTNAP5* for its enhancer function and retinal expression, we used post-GWAS prioritization. Multiple database search and analyses of available data indicated towards its putative role. Next, we used translation-blocking morpholinos to knock down *CNTNAP*5 in zebrafish to study its possible involvement in a functional consequence towards dysregulated retinal development. Subsequently, we performed immunofluorescence and western blot analyses using an acetylated tubulin antibody to observe the expression status and retinal neural tissue architecture. Furthermore, we examined the effects of *cntnap5* knockdown on apoptosis in zebrafish. In addition to immunofluorescence-based markers, we stained Nissl granules within neuronal cell bodies of retinal neurons to evaluate defects in ribosomal integrity reflective of neurodegeneration. Finally, in conjunction with the anatomical and histological analyses from genetic knockdown of *cntnap5*, we aimed at evaluating cntnap5 deficiency to precipitate measurable visual locomotory deficits equivalent to loss of vision in zebrafish.

## Materials and methods

### Ethics statement

This study was approved by the National Institute of Biomedical Genomics Institutional Animal Ethics Committee (NIBMG-IAEC) for all zebrafish related experiments under the certification number NIBMGZF/002/2022-23. NIBMG Zebrafish Facility is registered with the Committee for Control and Supervision of Experiments on Animals in India with a registration number 2194/GO/ReBi/S/22/CPCSEA. No sample from human participants were used in this study.

### Conditional analysis of *CNTNAP5* locus

We selected 13 Single Nucleotide Polymorphisms (SNPs) for analysis, each characterized by p-values, odds ratios (OR), and 95% confidence intervals (CI) from our previous GWAS on PACG (**S1 Table**). To determine if the association signal for glaucomatous neurodegeneration previously identified through GWAS points to *CNTNAP5* as the likely causal gene within the chromosome 2 locus, we performed conditional and joint analysis using summary statistics of SNP genotype-phenotype associations across the region. This was carried out using version 1.93 of genome-wide complex trait analysis (GCTA) software[14]. LD-independent genome-wide significant SNPs within ±500kb window flanking *CNTNAP5* transcription start site were selected from the broader chromosome 2 locus showing evidence of association in the primary GWAS discovery analysis. The GCTA stepwise model selection (—cojo-slct algorithm) was applied conditioning on the top sentinel SNP across iterations to refine association signals, pinpoint candidate independent causal variants, and infer local genes mapping to said variants statistically accounting for most of the GWAS signal.

### Analysis of Cis-Regulatory Elements (CRE) annotations at *CNTNAP5* locus

To investigate the non-coding regulatory landscape surrounding prioritized candidate gene *CNTNAP5* for follow-up functional study, we searched for annotated CREs within the broader ±500 kb flanking region using the SCREEN DNA Elements Explorer platform [15]. We specifically queried genomic coordinates chr2:125,083,095–125,084,384 spanning the *CNTNAP5* locus for reported CREs characterized by empirical DNaseI, CTCF, H3K4me3, and H3K27ac signals. Data on genomic location, biochemical epigenetic readouts, normal cell/tissue source

identifiers and other annotation metadata were systematically compiled for downstream analysis in R software (R, version 4.1.2) environment as described in Chakraborty et al, 2023 [16].

## Analysis of *CNTNAP5* SNPs in DeepSEA database and RegulomeDB

We selected 13 SNPs for analysis, each characterized by p-values, odds ratios (OR), and 95% confidence intervals (CI) from our previous GWAS on PACG. These SNPs were prioritized based on their significant associations with the risk of PACG.We input genomic coordinates and reference/alternate alleles for the 13 *CNTNAP5* intronic and 3' UTR SNPs into the Deep-SEA web server (http://deepsea.princeton.edu). Regulatory impacts of SNPs disrupting or creating binding sites and composite functional significance scores were compiled from DeepSEA multi-task predictions exploring the known and predicted regulatory potential of PACG-associated non-coding variants annotated to the *CNTNAP5* gene provides additional support linking the genetic association signals observed through orthogonal bioinformatics evidence [17]. To assess potential genetic regulation of *CNTNAP5*, we examined 13 identified SNPs in RegulomeDB. This database integrates various regulatory information including expression quantitative trait loci (eQTLs) and chromatin immunoprecipitation (ChIP-seq) data to predict regulatory potential of variants. SNPs were input and the resulting RegulomeDB scores were recorded, with lower scores indicating increased evidence for regulatory function.

## Expression quantitative trait loci (eQTL) Analysis

Further to elucidate the potential impact of 13 selected SNPs within the *CNTNAP5* gene on its expression levels in the retina, eQTL analysis was conducted to assess the association between genetic variants (SNPs) and gene expression levels in retinal samples. The average expression of the *CNTNAP5* gene, measured in Counts Per Million (CPM), was derived from transcriptomic datasets of Age-related Macular Degeneration (AMD) retinal samples. The analysis utilized QTLtools, which computes nominal p-values to identify potential eQTLs. The identified eQTLs were integrated with methylation QTLs (mQTLs) to better understand the regulatory impact of these SNPs on *CNTNAP5* expression [18].

## Analysis of ocular tissue expression patterns of *CNTNAP5*

To evaluate the expression patterns of *CNTNAP5* across ocular tissue types, we obtained RNA sequencing datasets from the eyeIntegration v2.12 (https://eyeintegration.nei.nih.gov/) and human eye transcriptomic atlas(https://www.eye-transcriptome.com/) encompassing tissues from the cornea, iris, lens, ciliary body, retinal pigment epithelium, choroid, central and peripheral retina.

## Exploring chromatin topology at *CNTNAP5* locus using 3D, 3DIV and HUGIn genome browser

To investigate potential long-range chromatin interactions connecting distal regulatory elements to prioritized candidate gene *CNTNAP5*, we utilized the 3D Genome Browser (http://3dgenome.org). We searched 3D Genome Browser encompassing the full *CNTNAP5* gene body along with ±500kb flanking regions. Interactive conformation plots centred on the *CNTNAP5* Transcription Start Site (TSS) were generated to identify significant long-range intra-TAD chromatin loop anchors that may harbour distal regulatory elements. By utilizing overlays with tissue-specific epigenomic annotations, this investigation of 3D genomic architecture maps surrounding *CNTNAP5* facilitates the identification of potential non-coding

SNPs identified by GWAS signals that may have an impact on *CNTNAP5* expression [19]. Additionally, we queried the same 13 SNPs of *CNTNAP5* in the 3DIV database [20] to retrieve a list of statistically significant distal chromatin interaction partners and their genomic annotations. The distance normalized interaction frequencies and bias removed interaction frequencies were noted for each SNP. Further, we utilized the HUGIn browser [21] to visualize and compare the observed versus expected read counts between the *CNTNAP5* gene locus and 13 SNPs of interest with distal chromatin regions in specifically in in the dorsolateral prefrontal cortex (DPLC) and neuronal progenitor cells (NPC). The raw observed and expected read counts were extracted for each SNP for descriptive statistical analysis. By leveraging these publicly available Hi-C data resources, we obtained processed long-range chromatin interaction data between our loci of interest and distal regulatory elements across a human neural cell and tissue types. The normalized interaction frequencies and raw read counts were utilized to characterize and compare the chromatin architecture surrounding the *CNTNAP5* and its 13 SNPs under study.

## Gene Co-expression analysis

To identify genes co-expressed with *CNTNAP5*, we utilized two online databases—STRING (https://string-db.org/) and GeneMANIA (https://genemania.org/). In STRING, we input *CNTNAP5* and selected Homo sapiens as the organism. The analysis was run with a high confidence level (0.700). In GeneMANIA, we similarly input *CNTNAP5* with Homo sapiens selected. The co-expression analysis was run with automatic weighting. The resulting list of co-expressed genes from both databases were compiled.

## Pathway enrichment analysis

The combined list of *CNTNAP5* co-expressed genes was used as a input into Enrichr[22] for pathway enrichment analysis. The analysis was run against the KEGG and Reactome pathway databases with the default settings. Significantly enriched pathways were identified using an adjusted p-value cutoff of 0.05.

## Analysis of *CNTNAP5* variant-phenotype associations using HumanBase

To examine *CNTNAP5*, the gene was searched in the HumanBase database(https://hb.flatironinstitute.org/). The algorithm returns confidence scores predicting the likelihood of association between the queried gene and various diseases. These confidence scores are derived from the tissue-specific gene networks and analysis of known disease ontology terms.

## Analysis of *CNTNAP5* variant-phenotype associations using PheWAS

We queried the database of Phenotype-Wide Association Studies (PheWAS) catalog (phewascatalog.org) [23]. This aggregator tool combines published genotype-phenotype association statistics across a diversity of disease states profiled in large genomic datasets like the UK Biobank. We searched for all SNPs and small insertions/deletions mapped to within the genomic co-ordinates of *CNTNAP5* on chromosome 2 (chr2:125,083,095–125,084,384). Statistical association data linked to clinical diagnosis billing codes (ICD-10 classifications) were compiled across available PheWAS studies for variants catalogued within this *CNTNAP5* locus. Significantly associated phenotypes and disorders (at FDR or Bonferroni adjusted p-values $< 0.05$ threshold) were systematically tabulated and categorized by broad human diseases.

## Dual luciferase reporter assay

To determine our PACG associated prioritized SNP rs2553628 located downstream of the *CNTNAP5* locus exerts regulatory effects on enhancer activity; we carried out dual luciferase reporter assay. A 245 bp region flanking the rs2553628 variant was PCR amplified from PACG patient genomic DNA and directionally cloned into the multiple cloning site upstream of the firefly luciferase cassette using KpnI and XhoI restriction enzymes (NEB) in the pGL4.20 luciferase construct driven by an SV40 promoter (Promega). Patient and control subjects' DNA were used to generate the risk and reference allelic reporter constructs. HEK293 cells were cultured in DMEM (Gibco) supplemented with 10% FBS (Gibco) and seeded into 24-well plates at a density of $5 \times 10^4$ cells per well. Cells were co-transfected after 24 hours with 50 ng of the *CNTNAP5* luciferase reporter constructs along with 10 ng of pRL-TK Renilla luciferase internal control using FuGENE HD transfection reagent (Promega). At 48 hours post-transfection, cell lysates were collected and Firefly and Renilla luciferase signals measured using the Dual-Luciferase Reporter Assay System (Promega) as per manufacturer's protocol on Tecan infinite 200 PRO (Tecan). Normalized promoter activity was calculated as the ratio of Firefly to Renilla luminescence. Differences in normalized luciferase ratios between SNP alleles across three biological replicates were statistically compared by Student's t-test.

## Zebrafish husbandry and maintenance

Zebrafish (*Danio rerio*) were maintained in recirculating Tecniplast benchtop rack systems under standard laboratory conditions on a 14-hour light/10-hour dark cycle. Adult fish were set up for natural spawning to obtain synchronized embryo clutches for experiments. Usually, zebrafish (Tubingen) begin breeding at first light (7 AM). After feeding at 6:30 PM on the preceding day, pairwise breeding was established on a sloped breeding tank filled with zebrafish water. In order to maintain a 2:1 female to male ratio, four females and two males were housed in the breeding tank. When the light came on at 7AM in the morning, they often breed the next day. Approximately 500 eggs were found at the base of the breeding tank. After utilizing a strainer to gather the eggs, the strainer was rinsed with 1X E3 media (5 mM NaCl, 0.17 mM KCl, 0.33 mM CaCl2, 0.33 mM MgSO4, and 0.01% methylene blue) before the eggs were put into the Petri dish (with an average of 25 embryos/Petri dish) and kept in the incubator at 28.4˚C. All experiments in this study were conducted in compliance with institutional animal protocols under Committee for Control and Supervision of Experiments on Animal (CPCSEA) approval (No. NIBMGZF/002/2022-23).

## Microinjection of morpholino

Translation blocker (TB) morpholinos and 5 base mismatch morpholino against *cntnap5a* and *cntnap5b* were purchased from Gene Tools, LLC. The *cntnap5a* and *cntnap5b* genes in zebrafish are orthologs of the human *CNTNAP5* gene, sharing significant sequence and functional similarity. Mismatched controls (MM) were designed with morpholino sequences that differ by five base pairs from the *cntnap5*-targeting morpholinos. These controls were used to ensure that any observed phenotypic differences, such as retinal disorganization, were specifically due to *cntnap5* knockdown rather than non-specific effects of the morpholino (MO) injection. MO and MM sequences are listed in **S2 Table**. Morpholino stocks were diluted in 10 mM KCl and 25% Phenol Red and each morpholino (2 ng) were mixed in a volume of 2 nL was microinjected injected into single cell-stage embryos using calibrated injection volumes [24]. S1 Fig represent the key steps of the experimental process including morpholino injection at the single cell stage, embryo development, eye measurements at 96 hpf, tissue processing, and subsequent histological and behavioural analyses.

## Eye morphological assessment

At 96 hpf (hour post fertilization), larval zebrafish were anesthetized in tricaine solution and lateral images captured under a stereomicroscope (LEICA M205 FA) for morphological measurements. Eye diameter, defined as the axial length between the nasal and temporal periphery across the centre, was quantified for each larva using LAS 4V.13 software (LEICA). A 2-sample t-test was performed to compare the gross anatomical eye size both vertical and horizontal diameter of morphants and mismatch control in zebrafish to evaluate changes in eye size and morphology resulting from *cntnap5* knockdown.

## Tissue processing and cryosectioning

96 hpf zebrafish were anesthetized in tricaine methane sulfonate and subsequently euthanized following approved animal care protocols (Ferguson et al, 2019) [25]. Zebrafish were fixed overnight at 4°C in 4% paraformaldehyde (PFA) in phosphate buffered saline (PBS). After fixation, samples were washed in PBS 2–3 times to remove residual PFA. Tissues were then immersed in 30% sucrose in PBS overnight at 4°C for cryoprotection. Subsequently, zebrafish were oriented in OCT compound in peel-away cryomolds and quickly frozen using dry ice. Cryosections were cut at 15 μm thickness using a cryostat (LEICA CM 1860 UV) and collected directly on charged slides (Autofrost, Cancer Diagnostics). Sections were adhered to the slides by drying overnight at room temperature and stored at -80°C until use. Prior to immunostaining, slides were thawed at room temperature for 30–60 minutes and dried completely before beginning staining procedures.

## Eosin and hematoxylin staining

The sections on glass slides were stained as follows: PBS for 12 mins, Mayer's hematoxylin (SRL) in aqueous solution for 15 sec, water for 15 sec, eosin (SRL) in ethanolic solution for 20 sec, followed by dehydration with 100% ethanol for 5 sec twice, and finally mounted in DPX (Sigma) with cover slip. The tissue sections were observed under the light microscope (EVOS XL core Invitrogen). Detailed retinal analysis was performed on a representative subset of histological sections from 6–8 larvae per injection batch for each group (control and morphant).

## Immunofluorescence staining and confocal microscopy

Immunofluorescence staining on cryosections was performed following previously published protocols [25]. Briefly, Cryosections were thawed and dried at room temperature for 30 minutes prior to immunostaining. Tissues were rehydrated with PBS and blocked for 1 hour at room temperature with 5% normal goat serum (Gibco) in PBS containing 0.3% Triton X-100. Sections were incubated with primary antibodies overnight at 4°C. The following primary antibodies and dilutions were used: mouse anti-acetylated tubulin (1:1000, Sigma Aldrich), rabbit anti-cntnap5 (1:200, Abgenex). After incubation, slides were washed 3 times with PBS and incubated with isotype-specific anti-rabbit secondary antibody labelled with Alexa Fluor 488 (1:500, Abcam) and anti-mouse secondary antibody labelled with Alexa Fluor 568 (1:500, Abcam) for 2 hours at room temperature protected from light. The nuclei were stained with 4′,6-diamidino-2-phenylindole (DAPI) (1 μg/mL, Invitrogen) for 5 min at room temperature in dark. Finally, sections were washed with 1X PBS and mounted with ProLong Gold Antifade Mountant (Invitrogen). Images were acquired using confocal microscopy (Nikon Ti2 Eclipse). Image analysis was conducted using Nikon NIS-Elements software and ROI intensity was compared by Student t-test using GraphPad Prism software.

## Generation of customized cntnap5 antibody for gene knockdown validation

The sequence of cntnap5 Protein (XP_009294486.1 5 isoform X2, *Danio rerio*) was analysed by Hydroplotter software from protein lounge (proteinlounge.com) to select the suitable epitopes for antibody generation. The selection is based on analysing the amino acid sequences in Kyte-Doolittle and Hopp-Woods plots and selecting a suitable hydrophilic region also Kolaskar Tongaonkar for antigenicity. Based on the above analysis we have selected RSERNV-REASLQVDQLPLR (571-589aa) for antibody development. (Antigenicity Index: 1.028, Hydrophobicity Index: -1.084, Hydrophilicity Index: 1.268)

A cysteine has been added in the N-terminus of the peptide for conjugation to carrier molecule. After synthesis of the peptide, it is conjugated to KLH (Keyhole lymphatic hemocyanin). Then it was immunized to the two healthy New Zealand white rabbits with freund's complete adjuvant (CFA) followed by freund's incomplete adjuvant (IFA). After four boosters the first immune sera were collected. The initial screening of the developed antisera was carried out by coating ELISA plate with the unconjugated peptide (200ng/well) (**S2A Fig**).

Based on the above information, Rabbit B 3rd bleed was selected for further peptide specific antibody purification by conjugating the free peptide with Sepharose 4B matrix. The purified antibody was again validated by employing indirect ELISA (**S2B Fig**).

## Whole mount immunofluorescence

At 32hpf, embryos were dechorionated and transferred in 4% 1x PBS. Samples were permeabilized in ice-cold 100% methanol drop wise at -20˚C for 2 hour and then washed with 1x PDT. Non-specific binding was blocked with 5% goat serum, 1% bovine serum albumin in PBS-Tween for 1 hr at room temperature. Embryos were incubated overnight at 4˚C with rabbit anti-active caspase-3 primary antibody (1:500; Cell Signalling Technology) to label apoptotic cells. On the following day, embryos were washed and incubated with Alexa Fluor 568 goat anti-rabbit IgG secondary antibody (1:500; Molecular Probes) for 2 hrs at room temperature. Samples were washed with 1x PDT prior to imaging. 4% methylcellulose was used for whole mount imaging under the confocal microscope [26].

## Neurotrace staining

Cryosections ranging from 12–15 μm thickness were collected on charged slides. Zebrafish eye tissue sections were stained with NeuroTrace 435/455 Blue Fluorescent Nissl Stain (1:50, Thermo Fisher) for 30 mins at room temperature as per manufacturer's instructions. Sections were then washed 3 times in PBS and mounted with ProLong Gold Antifade (Invitrogen). Images were captured using confocal microscope (Nikon Ti2 Eclipse). Image intensity was measured using Nikon NIS-Elements software. The whole eye was selected as region of interest (ROI) and the fluorescence intensity within the designated ROIs was quantified. In order to account for non-specific fluorescence, background correction was done when recording mean intensity values. ROI intensity was compared by Student's 2-sample t-test using GraphPad Prism software.

## Zebrafish protein isolation

Zebrafish embryos were collected for protein isolation at 96 hpf after morpholino injection. Embryos were washed with PBS three times to remove any E3 media remaining. Whole cell lysis buffer was added to the embryos (HEPES (20mM) pH 7.6, 100mM NaCl, 1.5mM MgCl$_2$, 1mM DTT, 0.1% Triton X-100, 20% Glycerol and freshly added 10ul 0.1 M PMSF, 5ul Protease

inhibitor cocktail per mL of buffer and homogenized using cordless mortar and pestle (Sigma-Aldrich) keeping on ice. After incubation in ice for 30 minutes the samples were centrifuged at 13,000 rpm for 20 minutes at 4˚C. The supernatant was collected and used for western blotting [27].

## Western blotting

Protein concentration of the prepared zebrafish protein lysates was estimated using Bradford reagent (Bio-Rad). Equal amounts of protein (30ug) were resolved by 10% sodium dodecyl sulphate–polyacrylamide gel electrophoresis. Proteins were transferred to nitrocellulose membrane. Membranes were blocked with 5% non-fat milk in TBST (Tris-buffered saline-Tween 20) for 1 hour and incubated with primary antibodies overnight at 4˚C. The following primary and secondary antibodies were used as per the mentioned dilutions: anti Acetyl-α-Tubulin Antibody (Sigma Aldrich), anti GAPDH (1:1000, CST), Goat Anti-Mouse IgG H&L (1:10,000, Abcam), Goat Anti-Rabbit IgG H&L (HRP) (1:10,000, Abcam). The blots were visualized using SuperSignal™ West Pico PLUS Chemiluminescent Substrate (Thermofisher) in Chemi-Doc XRS+ System (Bio-Rad). Densitometric quantification was measured in unsaturated images using ImageJ software (National Institutes of Health, Bethesda, MD, USA).

## Spontaneous movement analysis

Larval zebrafish at 96 hpf were arrayed into 6-well transparent microtiter plate (1 larvae per well and each time 3 well for mismatch controls and 3 well for morphants) containing 200 μL E3 medium in each well. Plates were loaded into the Zantiks MVP system for locomotor tracking. The system was calibrated to maintain constant 28˚C temperature and stabilize lighting for 30 minutes prior to recording. Following a 60-minute habituation period under darkness, the integrated software was programmed for automated alternating 10-minute epochs of light on and light off with complete darkness over a total 60-minute period. The movement tracks and total distance travelled (mm) per well were quantified for each light/dark interval period by the Zantiks tracking software. Statistical analysis was conducted using GraphPad Prism software (v9,). A nonparametric Mann-Whitney test was performed to compare the groups in light and dark separately.

## Results

### In silico and in vitro characterization of *CNTNAP5* genic region

Conditional analysis of thirteen genome-wide significant SNPs from our previous study clustered within a ~500kb region of *CNTNAP5* on chromosome 2 showed maintenance of statistically significant genomic association with PACG. This indicates *CNTNAP5* is likely a candidate gene for the glaucomatous trait (**Fig 1A**). Next, bokeh plot of linkage disequilibrium (LD) revealed that our earlier associated 13 variants of *CNTNAP5*, clustered into two LD block (**Fig 1B**). To contextualize the 13 PACG-associated intronic and downstream *CNTNAP5* variants, uncovered from conditional analysis in terms of their putative regulatory functions, we intersected their locations with candidate cis-regulatory elements (CREs) annotated in the region. Plotting the *CNTNAP5* SNPs relative to adjacent CRE genomic positions revealed two distinct spatial clusters exhibiting differences in predicted regulatory element abundance. One cluster (cluster I, Blue Square) containing rs17011381, rs2901264, rs2115890, rs733112, rs1430263, rs17011394 and rs780010) reside in a genomically sparse region largely devoid of annotated CREs. In contrast, the cluster II (Red square) encompassing rs17724018, rs2553625, rs17011420, rs2553628 and rs17011429 directly overlap with a dense hotspot of CREs enriched

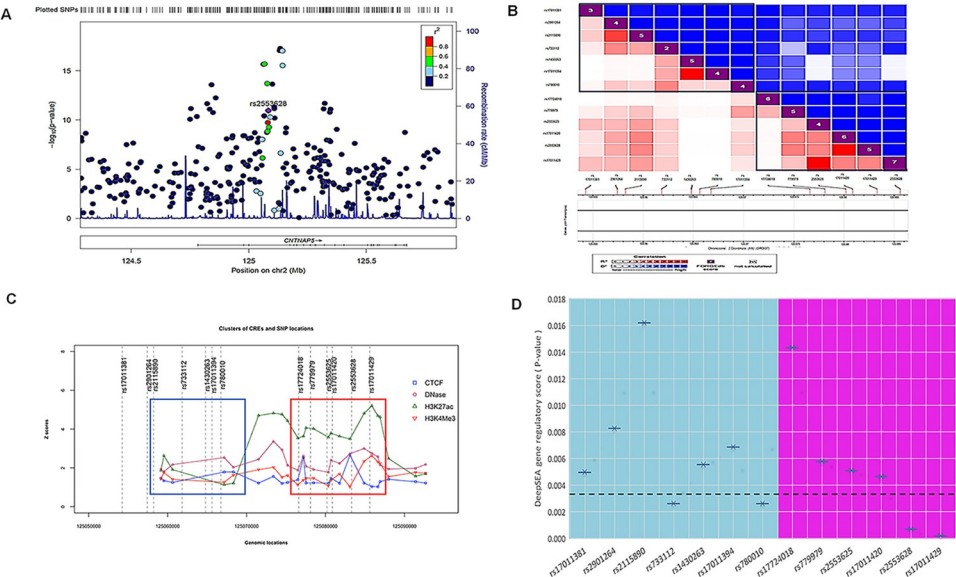

**Fig 1. A.** Regional Association Plot indicating the statistical strength of association among associated genomic region of *CNTNAP5* gene with PACG after performing conditional analysis **B.** Bokeh plot shows the pairwise linkage disequilibrium between the 13 variants around the region of *CNTNAP5* in two LD blocks **C.** CREs and SNPs plotted along with their genomic and epigenomic features of 13 SNPs of *CNTNAP5*. Based on the features, blue and red boxes are shown that roughly highlight SNP and CRE clusters. The elements and variants belonging to the red box show better likelihood for having functional relevance. **D.** The Whisker plot shows the DeepSEA gene regulatory score (P-value) of 13 SNPs of *CNTNAP5*. The plot includes a Bonferroni-corrected significance threshold line at 0.0038, indicating the adjusted significance level after correcting for multiple testing. SNPs below this threshold are considered statistically significant. Based on the features, blue (Cluster I) and pink (Cluster II) boxes are shown, representing different DeepSEA gene regulatory scores (P-values). The elements and variants belonging to the pink square have a better likelihood of having gene regulation potential.

in putative active chromatin marks (**Fig 1C**). This differential CRE density profile provides preliminary evidence suggesting the latter SNP cluster has a higher probability of harbouring non-coding variants interfering with functionally relevant gene regulatory elements that may contribute to a breakdown in *CNTNAP5* expression. Additionally, in DeepSEA database, we found same set (cluster II) of SNP are having significant enrichment scores on histone modifications demarcating putative active enhancer elements within introns of *CNTNAP*5 (**Fig 1D**). The RegulomeDB score of these 13 SNPs are enlisted in the **S3 Table**. Further the eQTL analysis of these SNPs within the *CNTNAP5* gene revealed varying degrees of association between genetic variants and gene expression in the retina. Among the significant findings, rs2553628 (p = 0.00529), rs17011420 (p = 0.00365) and rs17011429 (p = 0.00522, $r^2$ = 0.01930) showed a notable positive association with *CNTNAP5* expression, as indicated by their positive beta coefficients of 0.15117, 0.17926 and 0.31897, respectively. This suggests that the presence of these variants may enhance *CNTNAP5* expression levels. Additionally, rs17011381 (p = 0.0093) also displayed a significant positive association with a beta coefficient of 0.37025. Other variants such as rs2553625 (p = 0.04423) demonstrated more moderate effects. Conversely, SNPs like rs2901264 and rs2115890 had higher p-values (0.25371 and 0.10104, respectively), suggesting weaker associations with *CNTNAP5* expression. (**S4 Table**). The analysis of regulatory features for SNPs rs2553628 and rs17011429 revealed that both variants have high gene regulatory potential and both are residing in a strong linkage disequilibrium (LD). Examination of the HaploReg database showed that rs2553628 alters the motif for transcription factor TCF4 and having a CTCF binding site, while rs17011429 alters motifs for GR and Hsf.

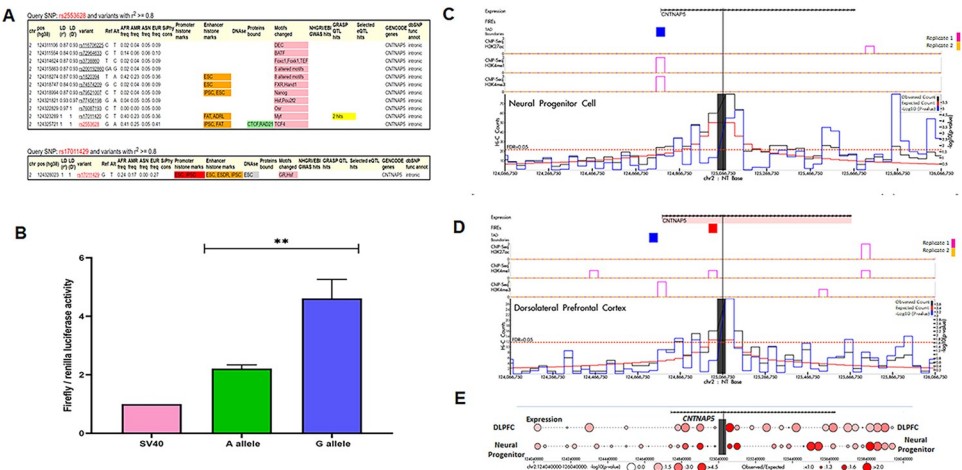

**Fig 2. A.** Variant detailed view of rs2553658 and rs17011429 in HaploReg. The regulatory motif TCF4 altered in reference and alternative allele of rs2553658. **B.** The data was drawn for a total of 3 biological replicates considering triplicate technical replicates for each biological experiments representing three different plasmid preparations (pGL3 SV40 minimal promoter vector) transfected in to HEK293T cells grown in 24-well plates. The height of the bar shows the mean value of all the 3 experiments, while error bars are depicting SE. The 4C plot of genomic region of CNTNAP5 (chr2:125,083,095–125,084,384) showing the FIREs, ChIP-Seq data and Hi-C counts in DPLC **(C)** and NPC **(D)** from HUGIn. The ratio of observed read counts to expected read counts is higher in NPC and is also statistically more significant than in DPLC **(E)**.

TCF4 is reportedly associated with neurological disorders [28]. Furthermore, rs2553628 demonstrated significantly altered motif activity between the reference and alternative alleles (Fig 2A). To functionally check the enhancer activity, in the dual luciferase assay, the G allele of rs2553628 (from the cluster II) was showing the significantly higher luciferase activity than the A allele (P value = 0.0034) (**Figs 2B** and **S3**). On further exploration on genomic architectural aspects surrounding *CNTNAP5*, we visualized Hi-C chromosomal conformation contact matrices encompassing the gene locus using the 3D Genome Browser. Notably, *CNTNAP5* resides within a topologically associating domain (TAD) suggesting capacity for long-range spatial contacts between the gene body and more distal enhancer elements (**S4A Fig**). The contact map, as well as the corresponding arc also suggests distal as well as proximal interactions in the regions surrounding the prioritized genic region in the dorsolateral prefrontal cortex (**S4B Fig**). Moreover, the HUGIn analysis of the genomic region of *CNTNAP5* reveals high proximal as well as distal interactions in the surrounding area as the observed counts were more than the expected counts especially outside the gene. The 4C plot from the database indicates the presence of frequently interacting regions (FIREs) near the *CNTNAP5* locus in the dorsolateral prefrontal cortex (DPLC), alongside peaks for H3K27ac, H3K4me1, and H3K4me3 in both DPLC and neuronal progenitor cells (NPC) (**Fig 2C–2E and S5 Table**). The Human Protein Atlas expression dataset showed that the expression of *CNTNAP5* is restricted only in retinal and neural tissues (**S5 Fig**). Analysis of human eye transcriptomic atlas datasets revealed marked enrichment of *CNTNAP5* expression within central retinal tissues amongst profiled ocular cell types (**S6 Fig**). The neural pathways were significantly enriched after performing pathways enrichment analysis of the co-expressed genes of *CNTNAP5* (**S7 and S8 Figs**). Analysis of the *CNTNAP5* gene in the HumanBase database revealed several disease associations. The strongest association was with autism spectrum disorder, with a confidence score of 0.54. A weaker association was also found with epilepsy (confidence score 0.11). Most notably, HumanBase analysis also predicted an association between *CNTNAP5* and retinal

disease, with a confidence score of 0.11. To visualize these results, the confidence scores for *CNTNAP5* across various neurological and retinal diseases were shown (**S6 Table**). Together, with the genetic and genomic architectural evidence, the expression patterns further strengthen the candidacy of *CNTNAP5* in contributing to retinal phenotype outcomes in PACG. Additionally, mining of the phenotype-wide association study (PheWAS) catalog revealed significant genetic associations between variants in the *CNTNAP5* locus and several neurological disorders including epilepsy, schizophrenia and Parkinson's disease (**S9 Fig**).

## *cntnap5* morpholino-injected embryos show ocular phenotypic alterations in zebrafish

Following extensive genetic fine-mapping data integration indicating *CNTNAP5* as a putative contributor in glaucomatous neurodegeneration, we next sought direct functional validation *in vivo* using antisense knockdown approaches in zebrafish larvae. Quantitative gross anatomical phenotyping revealed significantly reduced diameters along both the nasal-temporal axis (horizontal) and superior-inferior axis (vertical) of eyes measured from cntnap5 a and b translation-blocking double morpholino (dMO) injected embryos (n = 72, biological replicates = 6) at 72 hours post fertilization compared to 5 base pair mismatch (MM) morpholino controls (n = 72, biological replicates = 6) (P<0.001) (**Fig 3A and 3B**). There was no significant death observed in both control and morphant groups. The injection rate, death rate and other anatomical parameter was compared in both groups (**S10 Fig**). In deep phenotyping, histological examination showed noticeably diminished retinal nuclear layers with gaps and discontinuities in laminar organization in haematoxylin & eosin-stained thin sections from cntnap5 deficient larvae (**Fig 3C**).

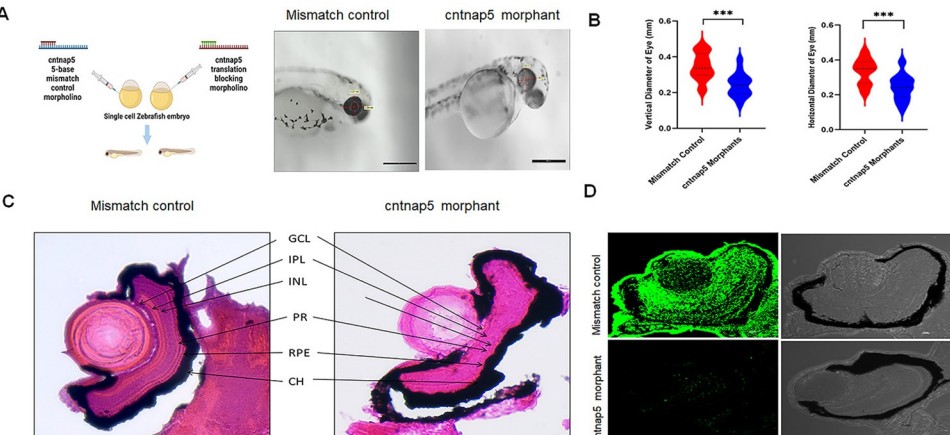

**Fig 3. A.** Morpholino knockdown of *cntnap5a* and *5b*, left cartoon is created with BioRender.com licensed version. Zebrafish were microinjected with a cntnap5 translation blocking morpholinos. Images taken 96 hpf, eye of a mismatch morphant (left) and an eye cntnap5 morpholino injected zebrafish (right) showing smaller anatomical eye than mismatch control. **B.** Quantitative gross anatomical diameters along both the nasal-temporal axis (horizontal) and dorsal-ventral axis (vertical) of eyes measured from cntnap5 translation-blocking morpholino injected embryos and mismatch morpholino controls zebrafish at 96 hours post fertilization stage. bars = mean ± SD, ns not significant, ***$p < 0.005$. **C.** Histological analysis of zebrafish eye microinjected with mismatch morpholino (left) a cntnap5 translation blocking morpholino (right) using H&E stain at 96 hpf stage showing disrupted retinal layers in cntnap5 morphant. GCL, ganglion cell layer; IPL, inner plexiform layer; INL, inner nuclear layer; PR, photoreceptor layer; RPE, retinal pigment epithelium; CH indicates choroid. **D.** Representative IF image of tissues stained with anti-cntnap5 showing cntnap5 expression in mismatch morphant (upper left; lower Transmitted Light Differential Interference Contrast (TD) image) but no expression in a cntnap5 translation blocking morphants (lower left: right: TD image).

## CNTNAP5 antibody mediated validation of gene knockdown

In conjunction with the gross morphological and histological analyses demonstrating perturbation of eye and retinal development upon repression of endogenous cntnap5 translation, we also aimed to directly confirm successful protein knockdown in our zebrafish model using a target-specific customized antibody. As expected, dMOs exhibited a near complete absence of detectable anti-cntnap5 immunofluorescence throughout the retinal layers of eye. In contrast, the mismatch controls show cntnap5 antibody immunostaining in their whole retinal layers, P value <0.0005 (**Figs 3D** and **S11**). Thus, our data clearly suggest that cntnap5 deficiency disrupts retinal development in larval zebrafish.

## *cntnap5* morpholino-injected embryo eye showed disturbed retinal cytoarchitecture

To investigate the retinal tissue architecture after deep phenotyping using H&E staining, the morphant retina showed profoundly increased anti-acetylated tubulin immunofluorescence signal intensity and disoriented retinal tissue organization throughout the retinal ganglion cell layer compared to mismatch morphant retina; P-value = 0.0108 (**S12 Fig**). To validate this, we co-stained with anti-cntnap5 antibody along with DAPI to stain the nuclei (**Fig 4A**). Similarly, western blot quantification of whole embryo lysates further corroborated a significant ~1.5-fold upregulation of acetylated alpha tubulin protein levels in cntnap5 deficient embryos relative to mismatch controls; P value = 0.031 (**Figs 4B** and **S13**).

## *cntnap5* morpholino-injected zebrafish embryos show visual neurodegeneration and greater apoptosis

During our investigation, we employed the NeuroTrace stain to examine zebrafish retinal neurons. Notably, our findings revealed a significant reduction in NeuroTrace stain mirroring less Nissil granule intensity of retinal neurons within the dMOs compared to the MMs (P<0.005)

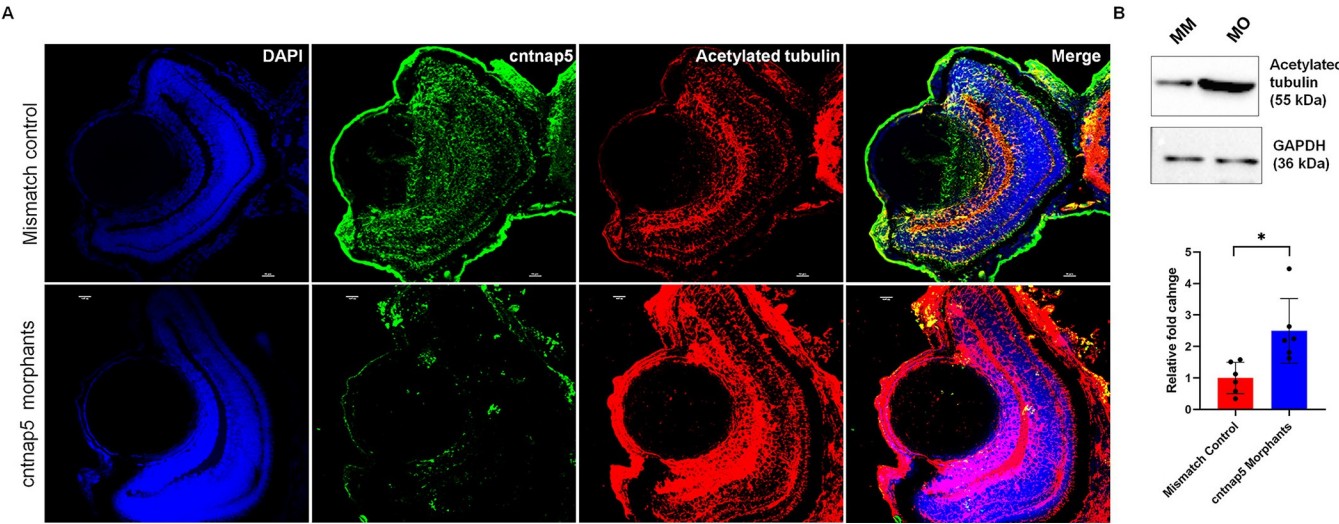

**Fig 4. A.** Cntnap5 and Acetylated Tubulin Expression in Zebrafish Eye Tissue (96 hpf): Representative confocal images of cntnap5 and acetylated tubulin expression of eye tissues from mismatch control fish (upper) and cntnap5 morphant (lower) zebrafish at 96 hpf (blue- DAPI, green- cntnap5, red- acetylated tubulin, and merged). **B.** Acetylated Alpha Tubulin Levels (Whole Embryo) (72 hpf): Western blot image of acetylated tubulin expression of whole zebrafish tissue lysate (72 hpf). MO: cntnap5 morpholino injected fish MM: mismatch control morpholino injected fish. A two-sided t-test. bars = mean ± SD, ns not significant, *p < 0.05.

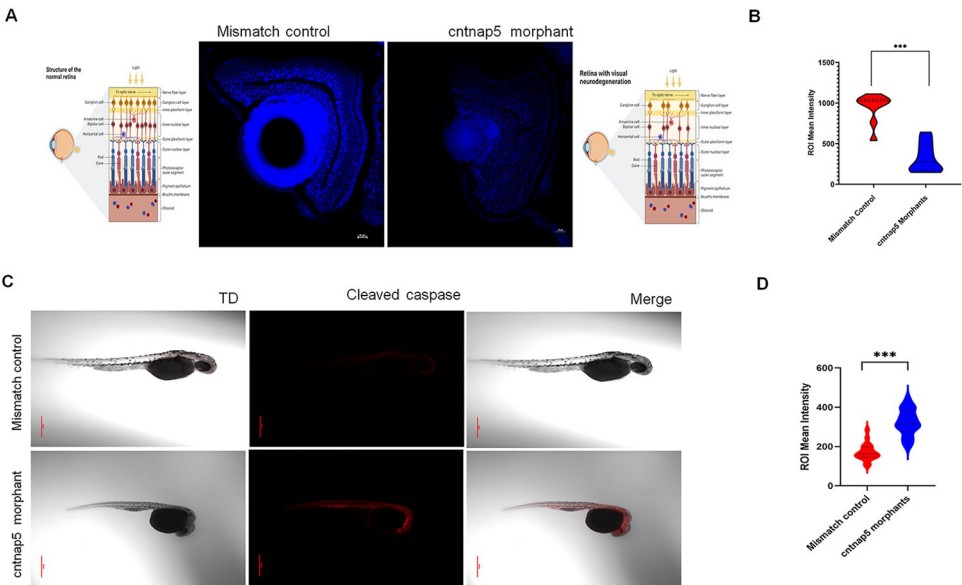

**Fig 5. A.** Representative eye tissue images of Nissl granules expression of zebrafish embryos (96 hpf) injected with a mismatch control (left) and cntnap5 morpholino (right). Corresponding cartoons are created using BioRender.com licensed version. **B.** Comparative analysis of eye mean intensity of eye for both groups. bars = mean ± SE, ns not significant, ***$p < 0.005$. **C.** Representative whole mount confocal images of cleaved expression of zebrafish embryos (32 hpf) injected with cntnap5 morpholino (upper) or mismatch control (lower) **D.** Comparative analysis of mean intensity of eye for the both groups, bars = mean ± SE, ns not significant, ***$p < 0.005$.

(**Fig 5A and 5B**). This observed decrease in NeuroTrace stain intensity strongly suggests the presence of neurodegeneration in the cntnap5 morphant zebrafish. Finally, we performed whole mount immunofluroscence to observe the apoptosis by cleaved caspase 3 as an executioner caspase at a system level. Whole mount cleaved caspase 3 immunofluorescence staining showed dMOs had significantly more cleaved caspase 3 staining compared to mismatch controls ($P<0.005$). This indicates greater apoptosis in the dMOs versus controls (**Fig 5C and 5D**). Together, these results demonstrate increased apoptosis in vivo in dMOs, supporting the importance of cntnap5 for cell survival.

## Spontaneous locomotory movement in *cntnap5* knockdown fish in light and dark phase

We analyzed the effects of *cntnap5* knockdown on visually guided locomotor activity under intermittent light/dark conditions using the Zantiks system. During 10-minute light periods, *cntnap5* morphants (dMO) showed significantly less average swimming distance compared to mismatch control larvae ($p<0.005$), reflecting impaired significant locomotory movement in bright light phase. However, distance moved in dark conditions were not significantly different between groups (p = 0.23) (**Fig 6A** and **S1 Video**). In summary, *cntnap5* morphant zebrafish spontaneous movement was impaired in light phase compared to mismatch controls indicating towards their vision loss (**Fig 6B**).

## Discussion

Our study highlights *CNTNAP5* as a novel gene associated with primary angle-closure glaucoma (PACG) with a potential role in retinal neurodegeneration. By integrating genetic fine-mapping, chromatin architecture analysis, gene regulatory annotations, and functional assays,

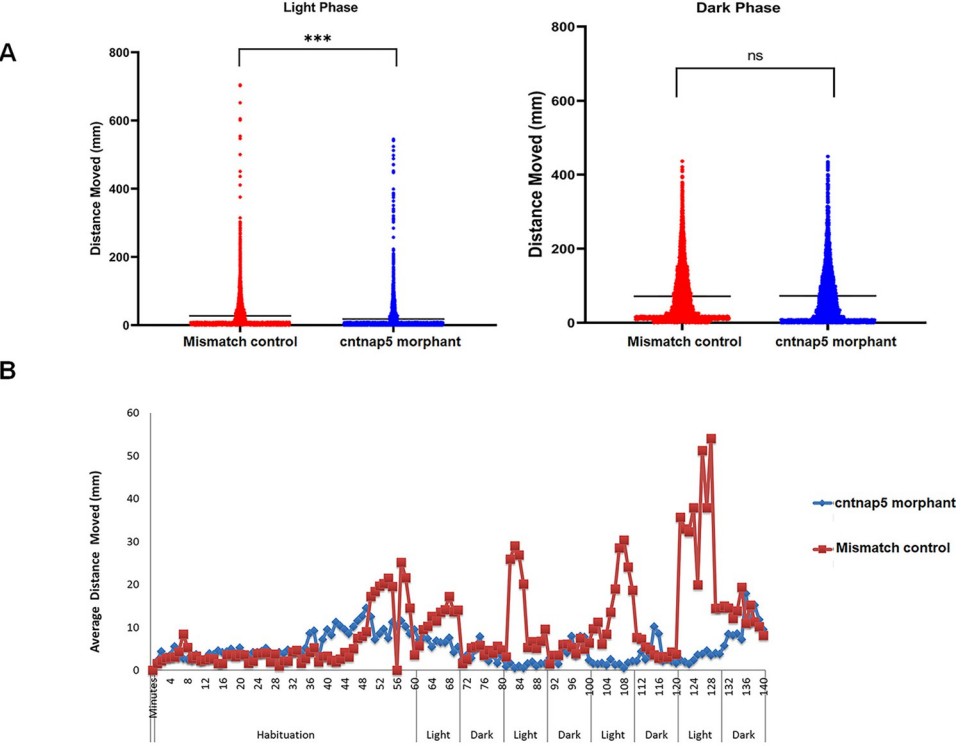

**Fig 6. Locomotory movement analysis of zebrafish. A**. Total distance moved (mm) of 96 hpf *cntnap5* morpholino injected zebrafish (blue circles) and 96 hpf *cntnap5* mismatch control (red circles) zebrafish for the light (left image) and light off (right image) periods. Left image, (light on) $p = <0.0001$, two-sided Mann–Whitney test, $n = 24$ *cntnap5* morpholino injected zebrafish and n = 24 *cntnap5* mismatch control, Right image (light off), $p = 0.5767$, bars = mean ± SEM ns not significant, $****p < 0.0001$. **B**. Average distance moved (mm) of 96 hpf *cntnap5* morpholino injected zebrafish (blue circles) and 5dpf *cntnap5* mismatch control (red circles) zebrafish exposed to four cycles of 10 min light on and 10 min light off after 60-minutes habituation period. Each point represents 1 min, $n = 24$ *cntnap5* mismatch control zebrafish and $n = 24$ *cntnap5* morpholino injected zebrafish.

we prioritized *CNTNAP5* as a strong candidate gene at the PACG risk locus on chromosome 2. Functional studies in a cntnap5-morphant zebrafish model further confirmed the role of *CNTNAP5* in maintaining retinal structure and function. Together, these findings suggest that *CNTNAP5* may contribute to the pathogenesis of glaucoma through its impact on retinal integrity and apoptosis.

## Broader implications for glaucoma genetics

Our findings shed light on the genetic landscape of PACG, particularly the role of *CNTNAP5* as a contributor to disease susceptibility. The genetic fine-mapping of the *CNTNAP5* locus identified several intronic and downstream variants that are significantly associated with PACG. These variants overlap with CREs, which are regions that regulate gene expression by modulating the interactions between promoters and enhancers. The presence of these regulatory variants suggests that changes in *CNTNAP5* expression levels could influence susceptibility to PACG, potentially altering the structural integrity of the retina.

The results of this eQTL analysis provide insights into how specific SNPs within the *CNTNAP5* gene might influence its expression levels in retinal tissues, potentially impacting retinal structure and function. The significant associations identified for rs2553628, rs17011420 and rs17011429 highlight their potential regulatory roles, as these variants appear

to increase *CNTNAP5* expression. This upregulation could be critical in understanding the broader role of *CNTNAP5* in retinal development and neurodegeneration.

The cross-referencing with the PheWAS catalogue revealed that *CNTNAP5* polymorphisms are not only associated with glaucoma but also with several other neurodevelopmental and neurological disorders, such as schizophrenia, bipolar disorder, epilepsy, and Alzheimer's disease [29–32]. This shared genetic basis indicates that *CNTNAP5* may be involved in a range of neural processes that influence both central and peripheral nervous systems, including retinal neurons. The expression of *CNTNAP5* in retinal cells, as observed in ocular tissue expression database, suggests that it may contribute to retinal development and maintenance, potentially making the retina more resilient to stressors like elevated intraocular pressure (IOP).

The role of *CNTNAP5* in regulating the interactions between promoters and enhancers suggests that disruptions in these regulatory networks could alter the expression patterns of genes involved in retinal integrity. Understanding the mechanisms through which *CNTNAP5* operate at the regulatory level could provide new insights into the genetic underpinnings of glaucomatous neurodegeneration and related conditions.

## Molecular Genetic Underpinnings of the Disease

Our study emphasizes the role of *CNTNAP5* in the regulation of retinal structure through its interaction with chromatin architecture and gene regulation. The Hi-C chromatin conformation data revealed that *CNTNAP5* is positioned within a topological associating domain (TAD), facilitating interactions between its promoter region and distal regulatory elements [33]. Additionally, the local density and activity of cis-regulatory elements (CREs) such as enhancers within a TAD can further modulate expression [34]. These interactions likely influence the expression of *CNTNAP5*, impacting its role in retinal development. The presence of PACG-associated variants within regulatory regions suggests that genetic variations may alter the structural organization of the retina by modifying *CNTNAP5* expression.

The dual luciferase reporter assays provided evidence of allele-specific enhancer activity for rs2553628, one of the variants within the *CNTNAP5* locus. The PACG risk allele rs2553628 showed increased enhancer activity, suggesting that this allele may promote higher *CNTNAP5* expression in certain contexts. Additionally, rs17011429, which has strong linkage disequilibrium with rs2553628, was identified as a co-eQTL associated with schizophrenia [29].This allele-specific regulation provides a potential mechanism by which genetic variations at the *CNTNAP5* locus could contribute to the development of retinal degeneration, by influencing *CNTNAP5* levels during critical stages of retinal development.

## In vivo analysis to support the hypothesized mechanisms of action

Using antisense knockdown experiments in zebrafish, we demonstrated cntnap5 knockdown results in ocular phenotypes including reduced eye size, disrupted retinal architecture with discontinuous nuclear layers as observed with reduced NeuroTrace intensity and increased acetylated tubulin levels. In contrast to the eye enlargement seen in human congenital glaucoma due to elevated intraocular pressure, the reduced vertical and horizontal eye diameters observed in *cntnap5*-morphant zebrafish likely reflect disruptions in retinal and ocular development due to *CNTNAP5* knockdown. Moreover, increased acetylated tubulin levels following cntnap5 knockdown indicate alterations in microtubule dynamics in cntnap5 morphant retinas possibly leading to cytoskeletal defects and disturbed cytoarchitecture. Acetylation of tubulin is a post-translational modification that alters microtubule stability and intracellular trafficking [35]. Earlier reports suggested that tubulin acetylation restricts axon over branching by reducing microtubule plus-end dynamics in nerves [36]. Defects in microtubule-based

transport could disrupt inner retinal laminar organization which manifest as gaps and discontinuous histology. Dysfunctional intracellular trafficking resulting from tubulin hyperacetylation provides a plausible mechanism contributing to the neurodegenerative effects of *cntnap5* knockdown. Axonal transport defects are linked to several neurodegenerative diseases [36,37]. Additionally, while the morphant eyes exhibited disorganization across all retinal layers, which contrasts with the selective retinal ganglion cell (RGC) loss typical in human glaucoma, this broader disorganization likely reflects the developmental role of *CNTNAP5* in maintaining retinal structure and cell integrity.

Our findings reveal a significant reduction in NeuroTrace stain intensity in the retinal neurons of *cntnap5* morphant zebrafish compared to mismatch control zebrafish. The decreased NeuroTrace stain intensity indicates loss of Nissl substance and suggests the presence of neurodegeneration in the morphant zebrafish retina. The loss of Nissl staining we observed in all the neuro-retinal layers of cntnap5 morphants reflects chromatolysis, or disintegration of Nissl bodies, which is a characteristic feature of neurodegeneration [38]. This indicates that *CNTNAP5* is essential for maintaining the structural integrity of retinal layers.

Our findings of increased cleaved caspase 3 immunofluorescence in the wholemount cntnap5-morphant zebrafish provides evidence that *cntnap5* deficiency leads to greater apoptosis during development. The caspase family of cysteine proteases play a central role in programmed cell death pathways [39]. Executioner caspases including caspase 3 are activated by initiator caspases in response to pro-apoptotic signals, ultimately leading to widespread proteolysis and cell death [40]. Increased cleaved caspase-3 staining in the *cntnap5* morphant retina indicates heightened apoptosis, suggesting cntnap5 is essential for retinal cell survival. This deficiency likely makes retinal neurons more vulnerable to apoptosis, leading to thinner retinal layers. Similar findings in glaucoma link disrupted pro-survival signalling to neurodegeneration through caspase-mediated cell death. Our data support a link between cntnap5 loss and increased retinal apoptosis, connecting this novel glaucoma risk gene to the apoptotic death of retinal neurons [40–43].

*CNTNAP5* belongs to the neurexin superfamily, a group of proteins known for their roles in synaptic organization, neural signalling, and cell adhesion [29]. Within the retina, these functions are essential for the proper arrangement of neurons and the maintenance of retinal structure. The role of *CNTNAP5* in axon guidance and synaptic stability likely contributes to its ability to maintain the alignment and structure of retinal cellular architecture, preventing structural abnormalities that could predispose the retina to further damage. Supporting these roles, our findings clearly show that knockdown of *cntnap5* in zebrafish larvae result in widespread retinal disorganization and overall disruption of cellular architecture of the retina.

Finally, our analysis of visually guided behaviour provides functional evidence that *cntnap5* knockdown impairs vision in larval zebrafish. Analysis of locomotor activity clearly showed significantly decreased movement during intermittent light periods in *cntnap5* morphants compared to controls. However, swimming distance in dark period was equivalent between groups. This selective reduction in light phase locomotion indicates the *cntnap5* deficient larvae have impaired visual function despite intact neuromuscular capacity. Similar light/dark dependent behavioural defects have been reported in zebrafish models of inherited blindness [44]. Visually-guided locomotion requires integration and transmission of visual cues to motor circuits [45]. Decreased photoperiod movement in *cntnap5* morphants reflects an inability to respond appropriately to light onset due to deficiencies in retinal processing. Loss of *cntnap5* expression likely impairs retinal circuit development and function, manifesting as vision-dependent locomotor deficits. Further electrophysiological analysis of visual responses and circuit connectivity in cntnap5 morphants will provide greater mechanistic detail into how *cntnap5* loss impacts retinal signalling pathways leading to functional vision loss.

However, our locomotor activity data clearly demonstrates *cntnap5* plays a critical role in the proper development of visual behaviour and function in zebrafish larvae. Our findings are consistent with the morphological and histological retinal abnormalities resulting from *cntnap5* knockdown underlying vision loss.

## Conclusion

This study has some limitations that should be considered when interpreting the identification and prioritization of *CNTNAP5* as a PACG risk gene. While Hi-C data suggested that *CNTNAP5* resides in a TAD, 3C-based sequencing was not performed, precluding direct measurement of chromatin looping interactions with risk variants [46]. Future studies involving 3C analysis and mammalian models could validate these findings and provide deeper insights into the regulatory mechanisms of *CNTNAP5* in glaucoma. Our study uses a zebrafish model to explore the role of *cntnap5*, initially identified through a human GWAS, in retinal development and neurodegeneration. While many pathways influenced by *CNTNAP5*, such as synaptic architecture and neural circuit organization, are conserved between zebrafish and humans, further validation in relevant human cells/tissues is necessary to fully understand the gene's role in human neurological conditions and diseases like glaucoma. We acknowledge certain limitations in modelling PACG, which typically manifests in middle-aged and older adults. The use of larval stages, while advantageous for genetic manipulation and visualization studies, may not fully capture age-related changes in anterior chamber dynamics and other pathophysiological features characteristic of PACG in humans. We also acknowledge that our study did not evaluate intraocular pressure (IOP) or damage to the anterior chamber angle, key features of PACG. Our focus was on exploring the role of *CNTNAP5* in maintaining retinal cytoarchitechture/organisation that is disrupted in PACG, rather than modelling the full clinical spectrum of PACG, including angle closure and IOP elevation. While the current study was designed to investigate *CNTNAP5* loss-of-function effects through morpholino knockdown, we recognize that future overexpression studies would provide more complementary insights.

In conclusion, our integrated genomic, expression, and functional analyses provide compelling evidence for *CNTNAP5* as a novel PACG risk gene conferring susceptibility through altered regulation in retinal cell types. The effects of *cntnap5* knockdown in zebrafish further validate its role in neurodegeneration. Given its neural-specificity and involvement in cytoskeletal dynamics, *CNTNAP5* represents a promising target for future investigation and therapeutic development for PACG and related neurodegenerative conditions. Further studies should explore how modulation of cntnap5 expression influences intracellular trafficking and survival signalling in retinal ganglion cells. Elucidating the precise molecular mechanisms through which dysregulated *CNTNAP5* contributes to PACG pathogenesis will yield crucial insights into disease aetiology while nominating strategies for targeted intervention.

## Supporting information

**S1 Table. The table presents the association between 13 SNPs of *CNTNAP5* and PACG, showing P-values, odds ratios (OR), and 95% confidence intervals (CI).**
(DOCX)

**S2 Table. Morpholino sequence of cntnap5a, cntnap5b and their respective mismatch 5 base mismatch control.**
(DOCX)

**S3 Table. RegulomeDB score of 13 SNPs of *CNTNAP5*.**
(DOCX)

**S4 Table. Summary of eQTL Analysis for 13 SNPs at the *CNTNAP5* Locus: Table represents the eQTL analysis results for 13 SNPs associated with *CNTNAP5* expression.** Each row includes the Variant ID, chromosome location (Chr), nominal p-value indicating the statistical significance of the association, r_squared (coefficient of determination) representing the proportion of variance explained by the variant, beta (slope) indicating the direction and magnitude of the effect on *CNTNAP5* expression, and slope_se representing the standard error of the beta estimate. Variants with lower nominal p-values suggest stronger evidence of association with *CNTNAP5* expression.
(DOCX)

**S5 Table. Observed and expected counts of the intronic SNPs of *CNTNAP5* found from HUGIn.** These SNPs that were distributed in two LD blocks (the SNPs in pink belong to one LD block and the ones in green belonged to the other LD block) and showed similar observed and expected read counts.
(DOCX)

**S6 Table. HumanBase portal showed *CNTNAP5* as the highest confidence score for autism spectrum disorder.** The gene also showed low but measurable confidence scores for epilepsy and retinal disease. Other neurodegenerative diseases like Parkinson's and Alzheimer's showed very low confidence scores.
(DOCX)

**S1 Fig. A flowchart diagram outlining the key steps of the experimental process: morpholino injection at the single-cell stage, followed by embryo development, eye measurements at 96 hours post fertilization (hpf), tissue processing, and subsequent histological and behavioral analyses.** Part of this figure is created with BioRender.com, a licenced version to users of the National Institute of Biomedical Genomics, India.
(TIF)

**S2 Fig. A.** First and Third immune sera of Rabbit A and B tested against the antigen coated at 200ng/well and at 1:5000 dilutions of primary Ab. Pre-immune sera were used as control in place of primary antibody. Plates read after 15 min of enzyme substrate reaction and the absorbance were measured at 405nm. **B.** Immune sera (at 1:5000) and Purified antibody (at 200ng/well) tested against the antigen coated at 200ng/well obtained a value of 2.228 and 2.452 respectively. Pre-immune sera were used as control in place of primary antibody. Plates read after 15 min of enzyme substrate reaction and the absorbance were measured at 405nm.
(JPG)

**S3 Fig. The data was drawn for a total of 3 biological replicates considering triplicate technical replicates for each biological experiments representing three different plasmid preparations (pGL3 SV40 minimal promoter vector) transfected in to HEK293T cells grown in 24-well plates.** The bar plots the mean value of all the 3 experiments, while error bars are depicting SD. Two tailed student t-test for independent means was used for calculating statistical significance; *p < 0.05, **p < 0.01; ***p < 0.001, ns- = not significant. The luciferase assay shows G allele of rs2553628(*CNTNAP5*) has a statistically significant higher firefly / renilla luciferase activity than the A allele of rs2553628 (*CNTNAP5*) for both forward orientation and reverse orientation. P-value: 0.0034 (Forward set) P-value: 0.8041 (Reverse set).
(TIF)

**S4 Fig. A**. The Hi-C heatmaps clearly show strong TAD correlation around genomic region chr2:125,083,095–125,084,384 of *CNTNAP5*. **B.** Visualization of significant long-range chromatin interactions centered on the genomic coordinates chr2:125,083,095–125,084,384 located

in the *CNTNAP5* intron. The figure is organized from top to bottom as follows: 1 Normalized Hi-C contact map with TAD annotations (blue triangles). 2 Identified chromatin interactions, where arcs represent long-range interactions between genomic regions. 3 RefSeq genes in the region. 4 Interaction frequency graphs: o Blue bar graph: Bias-removed interaction frequencies. o Magenta dots: Distance-normalized interaction frequencies. o Green threshold line: Interactions with blue bars above this line are considered significant (2-fold greater than the background). The x-axis represents the genomic position, with the center focusing on the *CNTNAP5* intron region. Proximal interactions are shown closer to the center, while distal interactions extend further from the center. This visualization helps identify potential regulatory interactions between the *CNTNAP5* intron and other genomic regions, both nearby (proximal) and far away (distal).
(JPG)

**S5 Fig. Expression of CNTNAP5 is restricted only in retina and neural tissues (*Source*: *Human Protein Atlas*).**
(JPG)

**S6 Fig. Expression profile of *CNTNAP5* in the different eye tissue (*Source*: *Human Eye Transcriptome Atlas*).**
(TIF)

**S7 Fig. Co-expressed gene of *CNTNAP5*.** Co-expression values as coex-z (a coex z values) is greater than 3 is significant it implies to an FDR of 0.1% also ruling out that they are randomly co-expressed.
(TIF)

**S8 Fig. The gene enrichment analysis of the co-expressed genes of *CNTNAP5* from the following databases-A.** GO Biological Process **B**. GO Cellular Components.
(JPG)

**S9 Fig. This plot illustrates the phenome-wide association study (PheWAS) results for the genetic variant of interest.** Each point represents a distinct phenotype, with the y-axis showing the -log10(p-value) of the association and different colours indicating various phenotype categories as per the PheWAS catalog. The red box highlights the neurological category, which includes diseases such as Parkinson's disease and autism. The blue box emphasizes the psychological disease category, featuring conditions like epilepsy and schizophrenia. These specific diseases (Parkinson's disease, epilepsy, and schizophrenia) are of particular relevance to the findings discussed in the main text of this paper. The plot demonstrates the wide-ranging phenotypic associations of the genetic variant, with a notable concentration of significant associations in neurological and psychological domains.
(JPG)

**S10 Fig. The bar plots represent the mean value of all the 8 experiment sets, while error bars are depicting SD.** Two tailed student *t*-test for independent means was used for calculating statistical significance; *$p < 0.05$, **$p < 0.01$; ***$p < 0.001$, ns- = not significant.
(TIF)

**S11 Fig.** Representative confocal images of cntnap5 expression of eye tissues from mismatch control fish **(A)** and cntnap5 morphant **(B)** zebrafish at 96 hpf. **C.** Comparative analysis of mean intensity of eye for the both groups, bars = mean ± SE, ns not significant, ****$p < 0.005$.
(TIF)

**S12 Fig.** Representative confocal images of acetylated tubulin expression of eye tissues from mismatch control fish **(A)** and cntnap5 morphant **(B)** zebrafish at 96 hpf. **C.** Comparative analysis of mean intensity of eye for the both groups, bars = mean ± SE, ns not significant, ***$p < 0.005$.
(TIF)

**S13 Fig. The whole blot of acetylated tubulin (55KDa) and GAPDH (36KDa) in three replicates.**
(JPG)

**S1 Video. Locomotory movement video of zebrafish.** The distance moved of 5dpf *CNTNAP5* mismatch control (upper 3 wells) zebrafish and 96hpf *CNTNAP5* morpholino injected zebrafish (lower 3 wells) exposed to four cycles of 10 min light on and 10 min light off period.
(AVI)

## Acknowledgments

We would like to express our sincere gratitude to Dr. Jayshree Advani and Dr. Anand Swaroop from the National Institutes of Health (NIH), National Eye Institute (NEI), and Dr. Nilanjan Chatterjee from Johns Hopkins University for their invaluable contribution in analyzing the eQTL data for 13 SNPs within the *CNTNAP5* gene and providing expression data from their retinal transcriptomic dataset.

## Author Contributions

**Conceptualization:** Sudipta Chakraborty, Samsiddhi Bhattacharjee, Mahua Maulik, Moulinath Acharya.

**Data curation:** Sudipta Chakraborty, Jyotishman Sarma, Shantanu Saha Roy, Moulinath Acharya.

**Formal analysis:** Sudipta Chakraborty, Jyotishman Sarma, Sukanya Mitra, Sayani Bagchi, Sankhadip Das.

**Funding acquisition:** Sudipta Chakraborty, Moulinath Acharya.

**Investigation:** Sudipta Chakraborty, Samsiddhi Bhattacharjee, Moulinath Acharya.

**Methodology:** Sudipta Chakraborty, Surajit Mahapatra, Mahua Maulik, Moulinath Acharya.

**Project administration:** Sudipta Chakraborty, Moulinath Acharya.

**Resources:** Sudipta Chakraborty, Shantanu Saha Roy, Mahua Maulik, Moulinath Acharya.

**Software:** Sudipta Chakraborty, Sreemoyee Saha, Samsiddhi Bhattacharjee.

**Supervision:** Samsiddhi Bhattacharjee, Mahua Maulik, Moulinath Acharya.

**Validation:** Sudipta Chakraborty, Jyotishman Sarma, Shantanu Saha Roy, Samsiddhi Bhattacharjee, Mahua Maulik, Moulinath Acharya.

**Visualization:** Sudipta Chakraborty, Samsiddhi Bhattacharjee, Mahua Maulik, Moulinath Acharya.

**Writing – original draft:** Sudipta Chakraborty, Jyotishman Sarma, Shantanu Saha Roy, Samsiddhi Bhattacharjee, Mahua Maulik, Moulinath Acharya.

**Writing – review & editing:** Sudipta Chakraborty, Samsiddhi Bhattacharjee, Mahua Maulik, Moulinath Acharya.

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
