## [Decision Letter · Decision Letter 0]

8 Oct 2024

Dear Dr Acharya,

Thank you very much for submitting your Research Article entitled 'Functional investigation suggests CNTNAP5 involvement in glaucomatous neurodegeneration obtained from a GWAS in primary angle closure glaucoma' to PLOS Genetics.

The manuscript was fully evaluated at the editorial level and by independent peer reviewers. The reviewers appreciated the attention to an important topic but identified some concerns that we ask you address in a revised manuscript.

We therefore ask you to modify the manuscript according to the review recommendations. Your revisions should address the specific points made by each reviewer.

To resubmit, log into your Editorial Manager account and select the option 'Revise Submission' in the 'Submissions Needing Revision' folder.

Yours sincerely,

Shefali Verma, Ph.D.

Guest Editor

PLOS Genetics

Giorgio Sirugo

Section Editor

PLOS Genetics

Reviewer's Responses to Questions

**Comments to the Authors:**

Reviewer #1: The authors have conducted a meticulous study evaluating the functional effects of a novel gene CNTNAP5 in a zebrafish model. They are able to prove anatomical and functional changes in the model but have not conclusively demonstrated glaucoma or primary angle closure glaucoma.

A few general comments are as follows,

• GWAS, SNP, TSS etc: full forms to be given at first instance of use.

• CNTNAP5 has been represented in uppercase and lowercase +/- italics throughout the manuscript. Does it signify a difference in meaning, if not, please maintain consistency.

• Cup disc ratio has been mentioned as a clinical hallmark of glaucomatous optic nerve damage, but neuro-retinal rim damage is the actual hallmark of glaucomatous damage.

• At what stage of the process was ‘Injection of morpholino’ done? The manuscript mentions it after ‘Gross eye measurement’ which doesn’t sound right. For someone from a non genetics background, the timeline of processing of zebrafish embryos would be unclear. Including a flowchart diagram would be a good idea.

• At 96 hours post fertilization, are the zebrafish embryos or larvae? (lines 313 and 335)

• Lines 417 to 420, did all morphant retinas show disorganization, were all 72 eyes thus tested? In comparison, were all mismatch control retinas well organized?

Comments specific to glaucoma.

• This study demonstrated reduced vertical and horizontal diameters of eyes of morphant zebrafish larvae. In contrast, in humans, in the presence of congenital glaucoma, the eye is usually enlarged. The vertical and horizontal diameters would be larger in glaucomatous eyes of a child compared to normal eyes.

• Additionally, the morphant eyes showed disorganization of all retinal layers, this is unlike glaucoma where the ganglion cell layer and its axons would be damaged.

• This study has demonstrated, “The reduction in 520 NeuroTrace intensity specifically in the outer nuclear layer of the retina indicates that photoreceptor neurons are particularly impacted by cntnap5 knockdown.” Photoreceptors are not primarily affected in glaucoma.

• The study has not demonstrated damage at the angle of the anterior chamber angle (as seen in PACG), nor has intraocular pressure been evaluated. At present, what is there to say that the morphant eyes have developed glaucoma and not some other retinal disease?

• Primary angle closure glaucoma is a disease of middle and older aged individuals. Could this study be more appropriate if evaluation was done on adult zebrafish?

Reviewer #2: Comments by section:

Figures:

-Quality and resolution of figures 1 and 2 is low

-Fig 1D unclear if SNP cluster II is statistically significantly different in p-values than cluster I - add labels to each cluster, add in Bonferroni significance dotted threshold line

-Fig S3B is a bit confusing, the first half of it does show the long interaction arcs, but then what regions we are looking at in the bottom half of the figure is unclear - distal and proximal interactions are not obvious - provide more context in the caption

-Missing closing parenthesis in fig S6 caption

-Cite phewas catalog when mentioning fig S8 and perhaps modify the figure to show the diseases within the neurological category or label the data points with the relevant diseases mentioned in the text of the paper (epilepsy, schizophrenia, and Parkinson’s disease)

-I would suggest titles mentioned either in the figures for clarity, ex. for fig 4B I would add something like “Acetylated Alpha Tubulin Levels (Whole Embryo)” for clarity

-Graphical abstract has resolution issues and portions of the figure are highly condensed (ex. structure of retina) or stretched out (ex. zebrafish larvae). Portions of the figure have spacing issues (ex. between the human icons in the PACS and PACG figure, size consistency of the arrows, etc.) Overall, this figure is unclear in listing each step.

Background + Methods:

-Not clear how we arrive at the 13 CNTNAP5 intronic and 3’UTR SNPs - these are the output from conditional analysis of the locus and the CRE analysis? Later sentence sounds like the 13 were identified in RegulomeDB. Unclear, please clarify. And if 13 SNPs were identified from the previous GWAS, please provide their information (odds ratio, p-value in a supplemental table)

-Have you considered overexpression analysis? Has this previously been done for cntnap5 and if so, what are the phenotypes?

-Presumably the previous GWAS from which CNTNAP5 was selected was performed in human data - please mention if the CNTNAP5 gene in zebrafish is orthologous with similar function as in humans or if the zebrafish was transgenic with a humanized version of the gene

-Previously cited association of CNTNAP5 with neurological conditions are mentioned but do we know what mechanisms (if any) this gene has been hypothesized to play a role in that leads to dysregulation and disease progression/development?

-Additional details behind the intuition of mismatched controls would be helpful as general knowledge for the reader.

Discussion:

-I find the discussion section difficult to read and parse as it is discussing many different results one after another. I would like to see a restructuring so that the main overarching points are discussed first, with the results mentioned as supporting details that can be further delved into. For example, having a zoomed-out view in the first part of the discussion will be helpful to orient the reader to the main aims and findings of the study. This can then be broken down into 1) the broader implications of revealing glaucoma genetics, 2) the molecular genetic underpinnings of the disease, and 3) finally the in vitro analysis to support the hypothesized mechanisms of action. (i.e. Line 501 onwards reads as though it could be the beginning of the discussion, giving us the larger picture and then breaking it down to specific biology.)

-I would also like to see more discussion of the results beyond stating the molecular interpretation of the individual result.

-You may consider adding headings into the discussion as well that can help organize the conversation based on different axes of analysis, and then also separate out the limitations and conclusions

-I would recommend including an overarching figure of a hypothesized mechanism of action; this would be very helpful for the reader to put together the rest of the analyses that were done to unveil different aspects of function. Please note that the graphical abstract does provide some level of overview, but additional text could be provided between the arrows to more clearly indicate the steps of the study.

-Missing period on line 546

Conclusion:

-Given that the target was identified in a GWAS, can you elaborate on how your results should be interpreted in the context of human biology? Are the pathways and mechanisms discussed consistent in humans as well or are these results to be viewed specifically with respect to zebrafish?

**Have all data underlying the figures and results presented in the manuscript been provided?**

Reviewer #1: Yes

Reviewer #2: Yes

PLOS authors have the option to publish the peer review history of their article (what does this mean?). If published, this will include your full peer review and any attached files.

Reviewer #1: No

Reviewer #2: No

---

## [Editor Report · Decision Letter 1]

14 Nov 2024

Dear Dr Acharya,

We are pleased to inform you that your manuscript entitled "Functional investigation suggests CNTNAP5 involvement in glaucomatous neurodegeneration obtained from a GWAS in primary angle closure glaucoma" has been editorially accepted for publication in PLOS Genetics. Congratulations!

Yours sincerely,

Shefali Verma, Ph.D.

Guest Editor

PLOS Genetics

Giorgio Sirugo

Section Editor

PLOS Genetics

Aimée Dudley

Editor-in-Chief

PLOS Genetics

Anne Goriely

Editor-in-Chief

PLOS Genetics

Comments from the reviewers (if applicable):

**Data Deposition**

http://datadryad.org/submit?journalID=pgenetics&manu=PGENETICS-D-24-00496R1

**Press Queries**

---

## [Editor Report · Acceptance letter]

27 Nov 2024

PGENETICS-D-24-00496R1 

Functional investigation suggests CNTNAP5 involvement in glaucomatous neurodegeneration obtained from a GWAS in primary angle closure glaucoma 

Dear Dr Acharya, 

We are pleased to inform you that your manuscript entitled "Functional investigation suggests CNTNAP5 involvement in glaucomatous neurodegeneration obtained from a GWAS in primary angle closure glaucoma" has been formally accepted for publication in PLOS Genetics! Your manuscript is now with our production department and you will be notified of the publication date in due course.

With kind regards,

Katalin Szabo

PLOS Genetics

On behalf of:
